



# Types of pulsating aurora: Comparison of model and EISCAT electron density observations

Fasil Tesema[1,2], Noora Partamies[1,2], Daniel K. Whiter[3], and Yasunobu Ogawa[4]

[1]The University Centre in Svalbard (UNIS), Norway
[2]Birkeland Centre for Space Science, University of Bergen, Norway
[3]School of Physics and Astronomy, University of Southampton, Southampton, UK
[4]National Institute of Polar Research, Japan

**Correspondence:** Fasil Tesema (fasil.tesema@unis.no)

**Abstract.** Energetic particle precipitation associated with pulsating aurora (PsA) can reach down to lower mesospheric altitude and deplete ozone. It is well documented that pulsating aurora is a common phenomenon during substorm recovery phases. This indicates that using magnetic indices to model the chemistry induced by PsA electrons could underestimate the energy deposition in the atmosphere. Integrating satellite measurements of precipitating electrons in models is considered to

be an alternative way to account for such underestimation. One way to do this is to test and validate existing ion chemistry models using integrated measurements from satellite and ground-based observations. By using satellite measurements, an average/typical spectrum of PsA electrons can be constructed and used as an input in models to study the effects of the energetic electrons in the atmosphere. In this study, we compare electron densities from EISCAT radars with auroral ion chemistry and the energetics model by using pulsating aurora spectra derived from POES satellites as an energy input for the model. We

found a good agreement between the model and EISCAT electron densities in the region dominated by patchy pulsating aurora. However, the magnitude of the observed electron densities suggests a significant difference in the flux of precipitating electrons for different pulsating aurora types (structures) observed.

## 1 Introduction

Pulsating aurora is a diffuse type of aurora with distinctive structures as arcs, bands, arc segments, and patches that are blinking

on and off independently with a period of few seconds (Royrvik and Davis, 1977; Yamamoto, 1988). The sizes of pulsating aurora range from 10 to 200 km horizontally, and 10 to 40 km vertically and usually occur at around 100 km altitude (McEwen et al., 1981; Jones et al., 2009; Hosokawa and Ogawa, 2015; Nishimura et al., 2020; Tesema et al., 2020b). Pulsating aurora is often observed after midnight, during the recovery phase of a substorm, and at the equatorward part of the auroral oval, (Lessard, 2012; Nishimura et al., 2020) and it can persist for more than 2 hours (Jones et al., 2011; Partamies et al., 2017;

Bland et al., 2019; Tesema et al., 2020a). However, substorm growth and expansion phase PsA (McKay et al., 2018) as well as afternoon PsA (Berkey, 1978) have also been reported.

The latitude of pulsating aurora can span a wide range, which depends on geomagnetic activity and local time. In general, PsA often observed between 56° and 77° degrees of magnetic latitude (Grono and Donovan, 2020; Oguti et al., 1981). During





the post-midnight period it is restricted between 60° and 70° magnetic latitude and in the morning sector it moves to higher
latitudes, between 65° and 75°. The source location of these regions maps to the magnetosphere between $4\,R_E$ and $15\,R_E$
(Grono and Donovan, 2020). PsA is very common with an occurrence rate of about 30% around magnetic midnight (Oguti
et al., 1981) and above 60% in the morning sector (Oguti et al., 1981; Bland et al., 2019)

The precipitating electrons' energy during pulsating aurora spans a wide range of magnitudes, predominantly between 10
and 200 keV (Miyoshi et al., 2015; Tesema et al., 2020a). However, electron energies as low as 1 keV have also been reported
(McEwen et al., 1981). PsA can consist of microbursts of relativistic electrons in the high energy tail of the precipitation, which
makes PsA an important magnetosphere–ionosphere (MI) coupling process in studying radiation belt dynamics (Miyoshi et al.,
2020). A significant number of studies have shown that the precipitation of PsA electrons is driven by wave-particle interactions
(Miyoshi et al., 2010; Nishimura et al., 2010, 2020; Kasahara et al., 2018). Recent studies further show that chorus waves play
an important role in pitch angle scattering of electrons over a wide range of energy during pulsating aurora (Nishimura et al.,
2010; Miyoshi et al., 2020). Electron cyclotron harmonic (ECH) waves are also a possible candidate in causing pulsating
aurora, especially at the lower end of the PsA energy spectrum (Fukizawa et al., 2018; Nishimura et al., 2020).

A recent study by Grono and Donovan (2018) categorize pulsating aurora into three different types in relation to their
structural stability and motion along the ionospheric convection. Salient and persistent structures moving along the ionospheric
convection belong to patchy pulsating aurora (PPA), and transient structures with no definite motion characterize amorphous
pulsating aurora (APA), which are the dominant PsA types. In addition, the third category, patchy aurora (PA), consists of
very persistent structure with limited pulsation at the patch edges. The energy of electrons associated with the pulsating aurora
types are different (Yang et al., 2019; Tesema et al., 2020b). From a total of 92 PsA events Tesema et al. (2020b) compared
the D region ionization level obtained by EISCAT radars for different types of PsA and suggested that PPA is the dominant
type of aurora affecting the D region atmosphere. The different categories of PsA reported in Grono and Donovan (2018)
originated from different source regions of the magnetosphere, where PPA and PA mapped entirely to the inner magnetosphere
while the APA source region spanned both inner and outer magnetosphere (Grono and Donovan, 2020). This indicates that
PsA can contribute to our understanding of the radiation belt dynamics as well, despite the challenges imposed by the large
spatio-temporal variation of the PsA structures.

Energetic PsA electrons can affect the chemistry of the mesosphere by strong production of odd hydrogen, which depletes
ozone in catalytic reactions (Turunen et al., 2016; Tesema et al., 2020a). As demonstrated by Tesema et al. (2020a) the softest
PsA precipitation does not have chemical consequences. It was further suggested in their study that it is mainly PA and PPA
that can most effectively ionize the atmosphere below 100 km.

In this study we test an ion chemistry and energetics model using measurements of precipitating electrons from low altitude
satellite as an energy input. We compared the EISCAT electron density measurements with the model output electron density to
investigate the ionization level during different types of pulsating aurora. This will enable us to understand the ionization rates
and energy spectra as they are measured at very different spatial and temporal resolutions, as well as the ionization changes in
the transitions between different PsA types.



## 2 Materials and methods

The optical data used in this study is from an all-sky camera (ASC) located in Tromsø (69.58°N, 19.21°E) in Norway, at the
same site with the EISCAT radars. It belongs to the network of Watec monochromatic Imagers (WMI) owned and is operated
by the National Institute of Polar Research (NIPR). The WMI consists of a highly sensitive Watec camera, a fish eye lens, and
band-pass filter at 428 nm, 558 nm, and 630 nm with bandwidth of 10 nm. The imaging system is capable of taking images
with 1 sec time resolution. In this study, we used images from the 558 nm filter. Technical details of the ASC can be found in
Ogawa et al. (2020).

POES satellite measurements are used to construct the spectrum of precipitating PsA electrons. The spectrum is used as an
input to the model. We approached the same procedure as explained in Tesema et al. (2020a) to construct the spectrum and
extrapolate the softer precipitation end using a power law function. This includes the energy range from 50 eV to 1 MeV.

Field aligned and vertical electron density measurements are obtained from VHF/UHF EISCAT radars located in Tromsø.
Instead of the standard 1 minute resolution data available for public on EISCAT database, we use a 5 seconds resolution
electron density processed using Grand Unified incoherent scatter design and analysis package (GUISDAP) software to match
with the high-resolution auroral imaging. The electron density measurements of the EISCAT radars are used to compare the
ionization level during pulsating aurora with the electron density from the model described below.

The auroral model used in this study is the combination of an electron transport code (Lummerzheim and Lilensten, 1994)
and a time-dependent ion chemistry and energetics model (Palmer, 1995; Lanchester et al., 2001), which solves the coupled
continuity equations for positive ions and minor neutrals above 80 km altitude.

In this study, we used a directly measured energy of precipitating electrons by POES satellite to construct the spectrum for
the input. We start the model run with an empty ionosphere, since prompt precipitation below 120 km does not respond to the
softer precipitation that is usually used to warm up the ionosphere for upper atmospheric studies. The run time and time step
for the model was about 3.5 seconds and 0.2 seconds, respectively. The minimum and maximum altitude of the model run is 80
and 500 km, respectively. Thus, the model does not reproduce ionization below 80 km, which corresponds to 100 keV (Turunen
et al., 2009).

The electron density output from the model is compared with the EISCAT-measured electron density. This will enable us to
answer the question of whether the overpass-averaged spectrum is a good representative as model input or if the patchiness of
the aurora should be considered in atmospheric models. Requiring the availability of EISCAT data, POES satellites overpass
and PsA from ASC images resulted in three events. Keograms (North–South overview) and ewograms (East–West overview)
of ASC images are constructed to further classify and study the pulsating aurora structures and the associated precipitation.

## 3 Results

Pulsating aurora can easily be identified from ASC keograms (e.g. Partamies et al., 2017), and categorized into different types
using ewograms (Grono and Donovan, 2018). A keogram is created by extracting north-south pixel columns of consecutive
individual all-sky images and stacking them in time and an ewogram is an east-west counterpart of a keogram. The energy




and flux of the precipitating electrons can be inferred indirectly from altitude and magnitude of the maximum electron density measured by ground-based incoherent scatter radars. Combining ASC data, EISCAT electron density measurements, electron density output from auroral model, and PsA energy spectra from POES measurements, we investigate the characteristics of precipitating PsA electrons and their ionization effects during three PsA events as follows:

## 3.1 Event 1: November 17, 2012

Figure 2 shows a keogram, ewogram, and EISCAT electron density measurements on November 17, 2012 between 4 and 5 UT. The keogram and ewogram are generated from one second time resolution ASC images taken at the Tromsø EISCAT site. Before 4:27 UT there was no electron density enhancement in the D and E regions, as there is no electron density enhancement nor auroral activity during this period. After 4:27 UT significant electron density enhancement (more than one order of magnitude) is seen below 110 km. Correspondingly, the ASC data showed PsA drifting into the EISCAT field of view (FOV) where it stayed until 4:43 UT. The PsA seen during this period is dominantly APA type. There is PPA type in the poleward region of the ASC FOV. After 4:43 UT this PPA drifted from north to east and became visible in the EISCAT radar FOV. The APA coverage started to diminish and the PPA took over most of the camera FOV. A clear transition in the EISCAT electron density is apparent at 4:43 UT. The electron density showed a thicker layer and precipitation reaching deeper, below 90 km, especially, after 4:49 UT. The thicker layer and more energetic precipitation corresponds to the PPA seen over the EISCAT radar.

Figure 3 shows the ASC images in 16 second intervals (a-f), the PsA spectrum constructed from POES measurements at the blue dots on the ASC images (g), electron density measured by EISCAT and modelled using the POES spectra (h-m), and green line emission intensity at the EISCAT (red) and POES (blue) measurement locations (n). From the ASC images it is clearly seen that the PsA structures are slowly drifting to east with decreasing intensity in the south (see also Supplementary Video one). This drift can be seen as patch lines (pathlines appear with or without stirations for patchy pulsating or patchy aurora, respectively (Grono et al., 2017)) in the ewogram on Figure 2. The median intensity of 10 pixels around the location of EISCAT (red) and the POES measurements (blue) are plotted in Figure 3(n). The intensity at the location of EISCAT over the entire duration was high, while at the location of the POES satellite measurements in the last three ASC images the intensity is extremely low. Looking at the electron density comparison between the model and EISCAT radar measurements, there is a good agreement between the two (Figure 3(h–k)) except the last two panels (Figure 3(l–m)), where the POES and EISCAT observations are looking into an entirely different region of auroral intensity. The first four points of POES observation spectra show similar magnitudes; curves (a–d) plotted in Figure 3(g) corresponding to the ASC observations in Figure 3(a–f). During this period, the altitude of maximum electron density in the EISCAT measurements was 95 km and from the model output it was 105 km. There is no significant differences in the electron density profiles as the FOV of EISCAT is mostly looking into a patch. The POES data points were also measurements within the patch "on" period, except in Figure 3(d), where there was very low emission (Figure 3(n)). Even though the emission intensity was low right after the whole FOV of the camera was filled with patches (as seen in the keogram plot on Figure 2), the electron density agreement between the model and EISCAT stayed the same in Figure 3(k). From Figure 3(d) POES is looking into a low emission region, which has correspondingly low fluxes in the spectra, which is similar to the spectra as in Figure 3(e–f). It is also clearly evident that above 10 keV the flux of





125 electrons in Figure 3(d) stayed similar to the previous three ASCs observations. However, for the last two ASC observations (Figure 3(e–f)) the POES observations were probably outside the precipitation region as the precipitating electron energies in the spectrum plot showed a large decrease above 10 keV on Figure 3(g). This causes a huge discrepancy between the model and EISCAT electron densities, accordingly.

  The spectra from POES (Figure 3(g)) does not show significant variations except for the last two spectra in time. Above 130 10 keV there is a significant drop in electron flux for the last two observations (Figure 3(e–f)). This corresponds to the low emission observed in those two points of the ASC images. The electron density comparison shows a good agreement between altitudes of 90 and 120 km in the first four panels. However, the last two panels show a big difference in the electron fluxes. The shape of the curves in these two panels are similar, and the gap between the curves below 80 km becomes narrower in these two panels.

  The altitude of maximum electron density showed a significant difference between the model run and the EISCAT observa-135 tions. However, the magnitude of electron density showed a good agreement between 85 and 120 km. The height of maximum electron density for the model output is about 105 km (corresponding to 10 keV electrons), and that of the EISCAT measurements 95 km (corresponding to 25 keV electrons) (Turunen et al., 2009). Note that the model can only reproduce electron density above 80 km, and thus, below 90 km the discrepancy between the two densities becomes large. The electron densities 140 above 120 km are due to the softer precipitation and were approximated by a power law function, which may not reproduce realistic ionization in this region. In addition, we did not perform warming up the ionosphere since we are interested in the prompt precipitation effects below 120 km. However, comparing the region between 85 and 120 km, the agreement between the model and EISCAT electron densities is good.

  The last two panels in the electron density (Figure 3(l–m)) comparison showed a kink-like structure at around 90 km, cor-145 responding to 40 keV electrons. From the spectrum it is apparent that above 40 keV the spectra for these two cases (magenta and cyan colors) showed almost the same fluxes. The median intensity around the EISCAT and satellite observations showed a large difference in these two panels (Figure 3(l-m)). From the EISCAT electron density plots shown in Figure 3(h–m), the zenith (black curve) and field aligned measurements (red curve) are similar. This event was studied by (Miyoshi et al., 2015) using the same EISCAT data, however, we used different ASC data, additional satellite data and model outputs in this study.

150 **3.2 Event 2: November 09, 2015**

Figure 4 shows keogram, ewogram, and EISCAT electron density measurements on November 09, 2015 between 2 and 3 UT. The keogram and ewogram are generated from one second time resolution ASC images in Tromsø. For this event a mixture of PsA types is clearly seen. Before 2:24 UT the PsA type was APA, which was followed by both APA and PPA (see also Supplementary Video two). During this one hour period the PsA structure and the magnitude of the electron density over the 155 ASC and EISCAT FOVs change significantly. After 2:24 UT the PPA starts to emerge from south and move northward to fill the FOV after 2:42 UT. The electron density significantly dropped between 2:04 and 2:28 UT (third panel of the Figure), when the EISCAT FOV was predominantly observing the APA type. After 2:44 UT the dominant PPA type corresponds to the increase





in electron density, and also deeper precipitation. It is also clearly seen that the width of the ionization layer starts to get thicker after 2:20 UT, when a mix of PsA types and later PPA is observed over the FOV of EISCAT.

Figure 5 shows the ASC observation, the POES spectra for the overpass data points (blue dots), electron density measurements at EISCAT (red dots in the ASC images) and electron density from the model output (blue curve) using the spectra obtained from POES (blue dots in the ASC images). The ASC images were dominated by two different auroral structures. The poleward portion of the ASC is filled with diffuse arc and the equatorward portion with patches. It is not clearly seen if the diffuse arc is pulsating or not. But displaying all images as a video (see supplementary material), the structure over the EISCAT

FOV is seen pulsating and can be categorized as APA. However, the POES measurements encounter a different type of PsA, namely PPA.

The spectra measured by POES are shown in Figure 5(e). The peak flux of electrons was observed below 10 keV. Above 100 keV, data point 4 showed significantly higher fluxes as compared to others. The height difference of the maximum electron density between the model output and EISCAT observations is small. However, the fluxes show more than one order of

magnitude difference. The emission intensity at data point 4 and at the EISCAT observation point showed a large difference. The POES data point 4 is entirely within the PPA precipitation region, while EISCAT is looking into the APA type. Note that this data point 4 showed higher fluxes in the energy range above 100 keV.

### 3.3   Event 3: January 13, 2016

Figure 6 shows the keogram, ewogram, and electron densities on January 13, 2016 between 5 and 6 UT. From the ASC, a very

slowly drifting and persistently stable structure of pulsating aurora is seen over the whole ASC FOV including the EISCAT FOV after 5:10 UT. A clear increase in electron density is observed when the pulsating patch is on and drifting in and out of the EISCAT FOV. The pulsating aurora over the entire FOV of the ASC is predominantly PPA during the one hour period, however, there are also some APA components seen in the keogram and ewogram plots. For example, before 5:15 UT APA type is seen in most of the ASC FOV. The ionization layer thickness also varies when the patch is visible in the EISCAT FOV

(see also Supplementary Video three). The thickness of the ionization around 5:25 UT is different from the thickness of the ionized layer seen just before 5:20 UT.

As shown in Figure 7(a-e) the POES satellite measurement is not co-located with the EISCAT location, however the structure of PsA is the same over the ASC FOV. As is shown in Figure 7(f), the POES energy spectra is very similar in magnitude and shape in all the overpassing data points. From the ASC images (Figure 7 (a-e)), the EISCAT is looking into the edge of a

pulsating patch, while the POES satellite is looking in to patch and the edge of a patch as it overpasses the pulsating aurora.

This event occurred very late in the morning, around 8 MLT. A persistent structure was observed over the whole FOV of the camera for the time period where the satellite is overpassing the region. The EISCAT electron density showed constant values at 95–115 km, but agrees well with the model electron density around 90 km. In Figure 7 (h-k), the electron densities showed a good agreement below 105 km. However, the discrepancy between the electron densities started at an altitude of 90 km (panel

g), and the electron densities below 87 km showed a significant difference (Figure 7(j-k)). There is no significant increase in the fluxes at any specific energy during this whole observation period, but the spectra rather showed a steep decrease at all the





energy levels. The median auroral emission intensity showed a similar decreasing trend at both the EISCAT location and at the satellite observation point.

## 4  Discussion

In this study, three PsA events were analyzed for their ionization characteristics. Each event analysis included high-resolution electron density measurements from the EISCAT Tromsø radar, high-resolution ASC images from the same site, and in-situ particle precipitation measurements from an overpassing POES satellite. The in-situ particle spectra were used as an input to an ionospheric model, and the model results were compared to the measured electron densities. Despite the differences between the space-borne and ground-based measurements, the conjugate measurements reveal some valuable details about the different

PsA types.

Event 1 on November 17 occurred very late in the morning sector, around 7:30 MLT, where harder precipitation is often reported to be present (Hosokawa and Ogawa, 2015). This is clearly seen in the EISCAT electron density measurements as a significant ionization below 80 km. Similar local time evolution of hardening precipitation was recently investigated by Tesema et al. (2020b) in a more statistical approach including EISCAT and optical data from the same geographical area. However,

the cut-off altitude of the model is 80 km, which causes a large discrepancy between the model and the EISCAT electron density below 80 km. The January 13 event (Event 3), which occurred about 30 minutes later in local time as compared to Event 1, showed a very good agreement between measured and modelled electron densities in the altitude range below 95 km during a softer type of precipitation. This indicates that the model is capable of reproducing measured electron densities very well within the height region of the prompt ionization at 80–120 km and during precipitation that primarily includes particle

energies which impact this height range ($\sim$1–100 keV). Our conclusions are thus focused on interpreting the height range of 80–120 km.

In two of the three cases (both November events 1 & 2) presented in this work, the PsA category changed within the observed one hour time period. During both events APA that was observed first changed into more persistent PPA. An enhancement in the measured electron density was observed at the same time with the optical transition between the two categories. Furthermore,

in the November 2015 event (Event 2) the POES satellite passed over the ASC station at the time of the transition between APA and PPA types. This resulted in a big difference, more than one order of magnitude difference between the electron densities and a difference in the altitude of the modelled and measured maximum electron density. In this Event 2, the satellite measured primarily the PPA type precipitation, while the EISCAT radar was looking mainly into the APA type precipitation. This suggests that a mixture of PsA types is the likely cause of the observed discrepancy.

As previously shown by a statistical analysis of PsA type Grono and Donovan (2020), APA has a tendency to occur at earlier local times than PPA and PA, i.e. around and even prior to midnight. A similar order of the PsA types was found in the two of our three case studies which included the transition between the different PsA types. This further suggests that the APA type may dominate the PsA events, which occur during (or in between) substorm activity and predominantly undergo increase patch sizes during the event evolution (Partamies et al., 2019). Because these PsA events are embedded into substorm aurora, and



thus would typically cover limited spatial regions as compared to PA and PPA, an overpassing spacecraft is likely to measure a mixture of different precipitation types and thus provide a false estimate for electron spectra at a near-conjugate ground location. However, deeper into the morning sector where the PsA is more often PA or PPA (Grono and Donovan, 2020) the regions covered by PsA are large. In this kind of case, our findings (clearest for Event 1) suggest that the overpass average of the in-situ particle spectra agrees well with the ground-based measurements of electron densities. As the overpass-averaged

spacecraft spectrum would necessarily include precipitation information for patches both in their on and off phases, this finding indicates that the patchiness of PsA is not a key factor in the energy deposition to the atmosphere. More detailed analysis is needed for a large number of different PsA types to confirm this result, but nonetheless this finding may have important implications for PsA modelling studies for atmospheric chemistry impact.

## 5 Conclusions

By combining EISCAT electron density, electron precipitation measurements from POES, and model electron density outputs, we study three PsA events identified using Tromsø high-resolution ASC data. We observed different types of PsA in the three cases. We showed that the near midnight PsA event (Event 2), which includes a mix of PsA types (APA and PPA), showed a significant electron density magnitude difference between EISCAT and model outputs. The model and EISCAT electron density magnitude in the morning sector events (Events 1 and 3), which consisted of measurements when the POES satellite

overpassed entirely over PPA types, showed a very good agreement. This suggests that the PsA spectra from POES used in modelling during a mix of PsA types could give an incorrect estimate if averaged spectra are used to model the energy deposition. However, the agreement during both the morning sector events indicated that overpassed averaged spectra are a very good estimate to model PsA energy deposition without considering the patchiness of the PsA. This also indicates that MLT dependence of PsA types might play an important role in future studies of atmospheric effects of PsA.

*Data availability.* The quicklook ASC images and keograms for event selection are available at Auroral Quicklook Viewer of NIPR ground-based network (http://pc115.seg20.nipr.ac.jp/www/AQVN/index.html) (last access: 26 February 2021). All-sky camera data are obtained by requesting the Principal Investigator of the auroral observation (uapdata@nipr.ac.jp) at National Institute of Polar Research (NIPR). Raw EISCAT data used in this analysis is available at http://portal.eiscat.se/schedule/schedule.cgi (last access: 26 January 2021) and GUISDAP software used to analyse the EISCAT raw data in high time resolution is available at https://eiscat.se/scientist/user-documentation/guisdap-

9-0/ (last access: 26 January 2021).

*Author contributions.* All authors contribute by providing necessary data, discussions and writing the paper.

*Competing interests.* The authors declare that no competing interests are present.

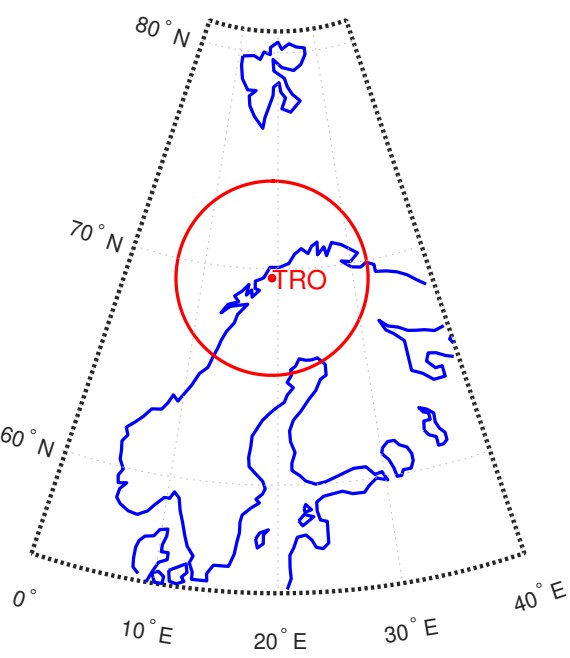

**Figure 1.** Geographic locations of ground-based ASC station and EISCAT radars in Tromsø (TRO) (red dot). The red circle marks the ASC FOV at about 110 km altitude. POES overpasses were selected so that their foot-points mapped to the ASC FOV.

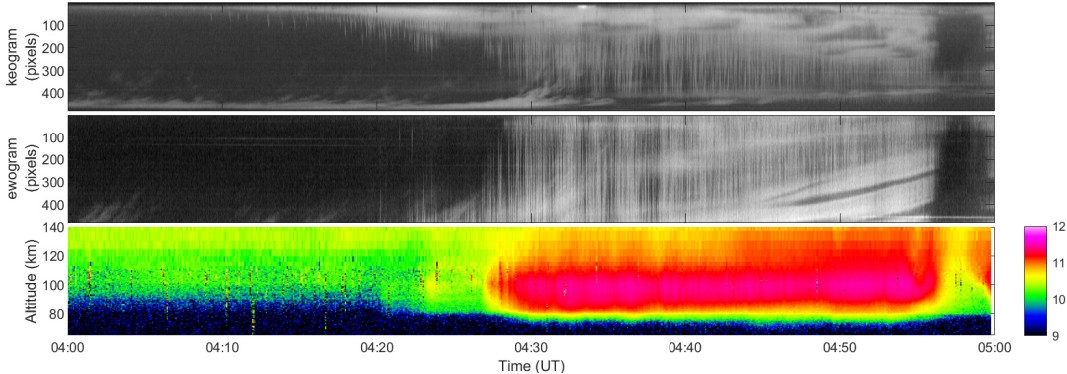

**Figure 2.** Keogram (top), ewogram (middle), and EISCAT electron density as a function of height (bottom) from UHF radar in Tromsø on November 17, 2012 between 4 and 5 UT. EISCAT beam points to the centre of the keogram at 235 pixel and in the ewogram at 245 pixel. The electron density is displayed in a logarithmic color scale.

*Acknowledgements.* The funding support for Fasil Tesema is provided by the Norwegian Research Council (NRC) under CoE contract 223252. In addition, the work of Noora Partamies is supported by NRC project 287427.





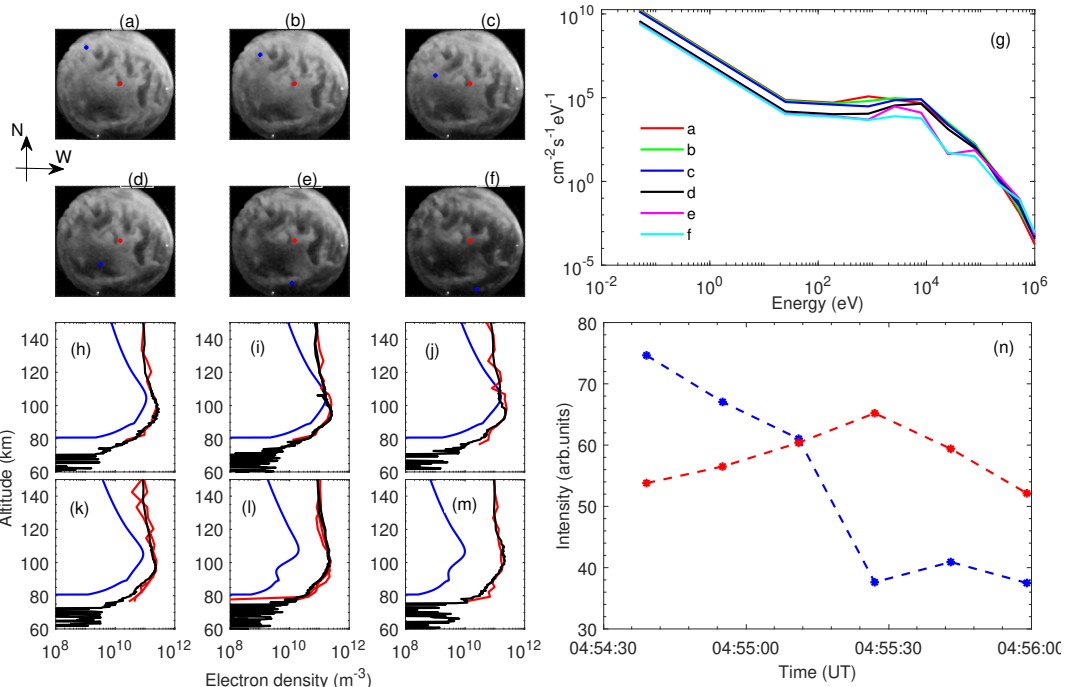

**Figure 3.** ASC images (a–f), spectra constructed from POES and power law extrapolation (g), curves labeled as a–f are corresponding spectra to the blue point on the ASC images, model and EISCAT electron densities (field aligned from UHF radar (red) and zenith measurements from VHF radar (black)) ((h–m) corresponding to the 6 ASC image times (a–f)), and relative auroral intensities at the location of satellite measurements as a function of time (n), blue dots at POES data points corresponding to (a–f) and red dots at EISCAT.

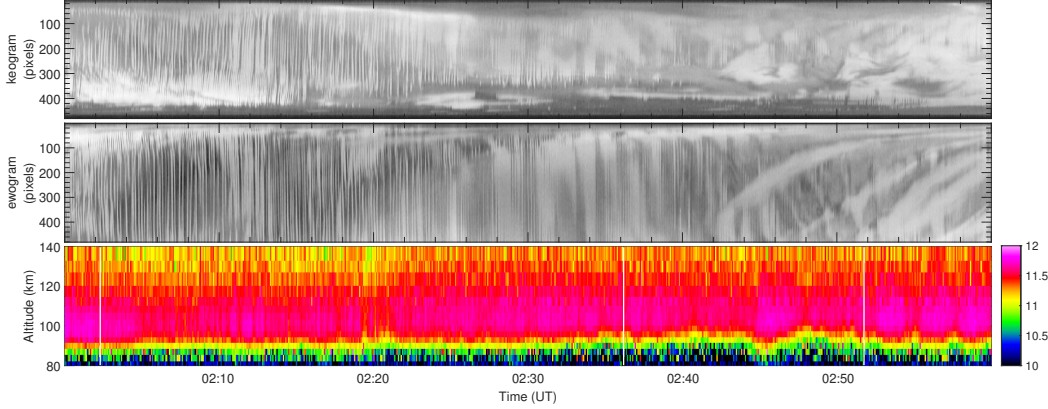

**Figure 4.** Keogram (top), ewogram (middle), and EISCAT electron density (bottom) from UHF radar at Tromsø on November 09, 2015 between 2 and 3 UT. EISCAT beam points to the centre of the keogram at 235 pixel and in the ewogram at 230 pixel. The electron density is displayed in a logarithmic color scale.





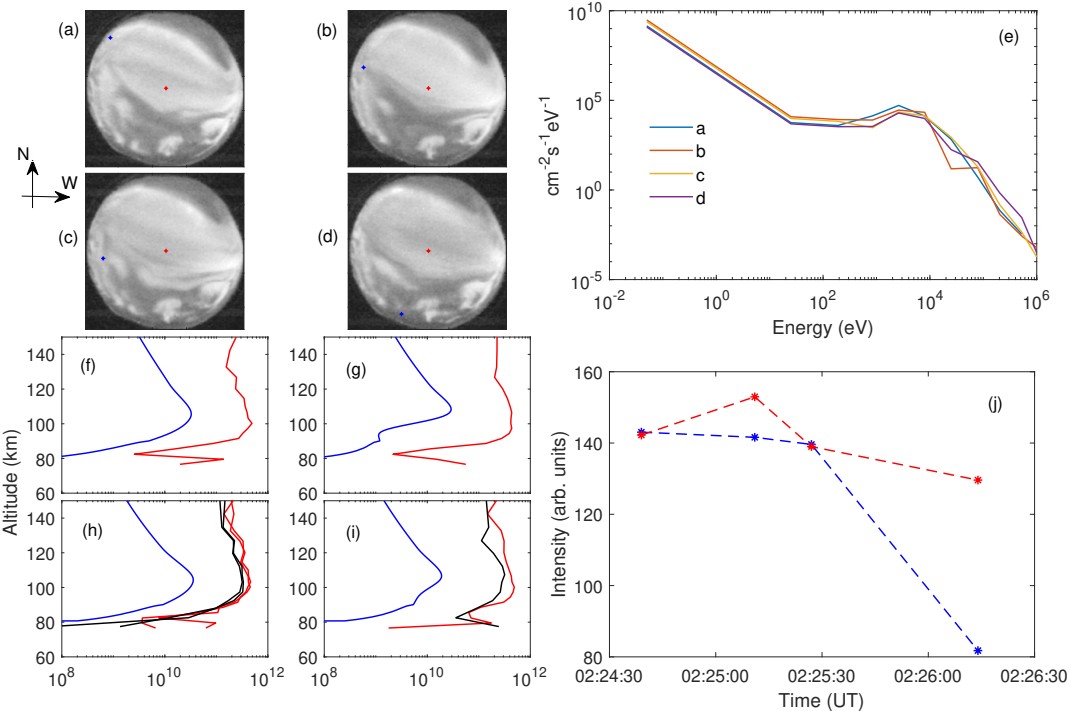

**Figure 5.** ASC images (a–d), spectra constructed from POES and power law extrapolation (e), curves labeled as a–d are corresponding spectra to the blue point on the ASC images, model and EISCAT electron densities ((f–i), colors as in Figure 3), and relative auroral intensities at the location of satellite measurements (blue dots) and at the EISCAT beam points (red dots).

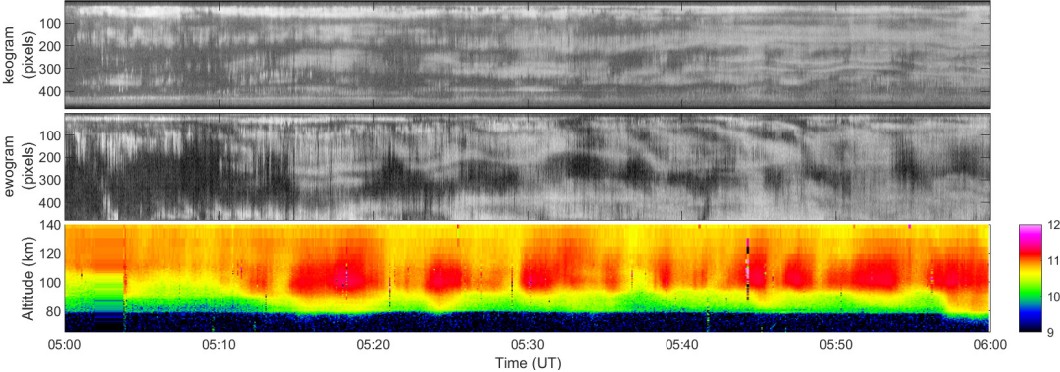

**Figure 6.** As in Figure 4 but for Event 3.





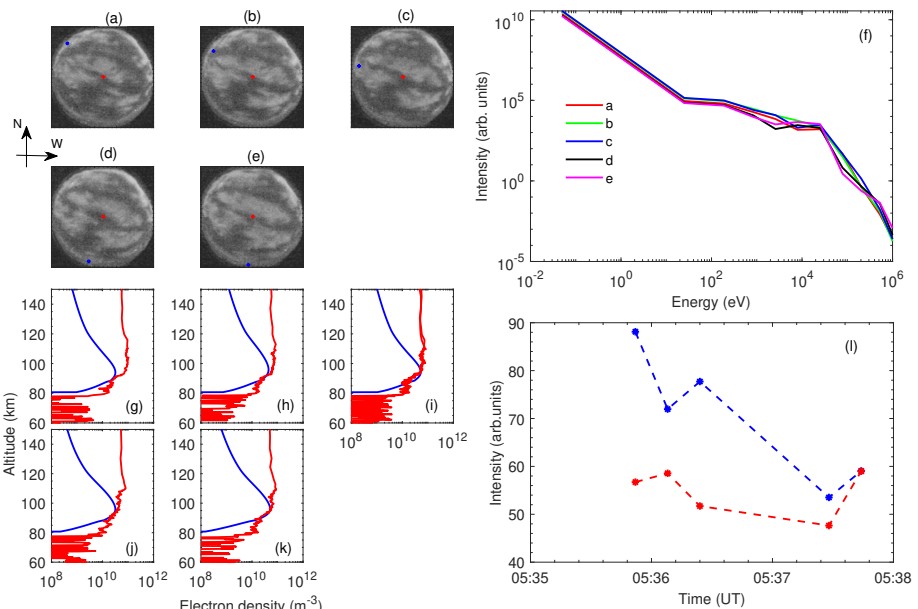

**Figure 7.** As in Figures 3 and 5 but for Event 3.

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
