# Peer review of "Types of pulsating aurora: Comparison of model and EISCAT electron density observations"

_Annales Geophysicae, 2021_

## Author Response (AR1)

**Referee #1**

We thank the reviewer for evaluating the manuscript and forwarding valuable comments. Suggestions and comments are included in the revised manuscript and the responses to the comments and suggestions are listed below (in italics):

L110: "stirations" spelling error.

- *Spelling error corrected*

L141: What is meant by "warming up the ionosphere"? Please explain.

- *Texts added to clarify this ( line 142-143)*

L143: "agreement...is good" is highly subjective. What is the definition of "good"? Looking at the plots suggests that "good" is half an order of magnitude. Please define "good" and state what the average difference is of the comparison over the height range of interest.

- *We accept the suggestion and correct is through out the manuscript (line 144)*

L180: "ionization around" should be "ionization layer around".

- *Corrected (line 181)*

L185: "in to patch" should be "in to a patch".

- *Corrected (line 186 )*

L209/21: "very well", please see comment on L143.

- *Corrected (line 209-210)*

L240: "very good agreement", please comment on L143.

- *Corrected (line 242)*

Figure 1: This figure is never introduced in the text. Please do so, or remove it.

- *Figure introduced in the text (line 60)*

Figures 2, 4 & 6: Please include units on the colour bars.

- *Units included on the color bars*

Figures 3, 5 & 7: The blue dot is hard or impossible to see in the images. Please use something clearer, e.g. yellow dot or cross.

- *Corrected by increasing the size of the dots*

Figure 3 caption: "model and EISCAT" should be "model (blue) and EISCAT".

- *Text added on the caption*

Figure 5 caption: "relative auroral intensities" should be "relative auroral intensities (j)" and the end of this sentence should indicate that the red and blue dots can be found in panels "(a-d)".

- *Text added on the caption*

**Referee #2**

We thank the reviewer for forwarding the comments. We include the comments, suggestions by adding texts and modifying figures. The responses to the comments are listed below (in italics).

Figures 2, 4, 6: Recommend putting vertical lines overlaid on these plots and indicating the regions of APA vs. PPA. It is hard to refer back and forth to the text to see when one type transitions to the other.

- *We added vertical lines on Figures mentioned and add texts in the caption of the Figures to describe the vertical lines.*

Throughout: Several times the ionization layer is referred to as expanding or increasing, but it would be worth the time to put this in more quantitative terms. Perhaps if you use a set density threshold, you can note when the threshold exceeded that density at various altitudes (e.g. the density increased above the threshold at 80 km after XX:XX UT) to give a better picture of exactly what altitude ranges these features span.

- *Instead of adding threshold electron densities we describe in terms of how much order of magnitude difference was observed in the electron density. On the figure, we also added a vertical line to mark the transition between PsA types.*

Line 213-214: Related to the last two comments, this line describes "an enhancement" at the time of "optical transition between the two categories." It would be much more clear if that transition time were defined clearly in the figure and a more precise measure of the enhancement was given.

- *We added texts to clarify see line 215-216.*

---

## Author Response (AR2)

We thank the editor for providing valuable comments. Corrections are included in the revised manuscript and the responses are listed below.

Consider changing the title of the second chapter "Materials and methods" to "Data and methods"

- "Materials and methods" is changed into "Data and methods" (Page 3)

Modifying/explaining the statement on page 3 and newly added one in the page 5 (paragraph 140) "… (the model run started from empty ionosphere)…". Please, note that, in general, the ionosphere is never "empty".

- Text added to clarify the statement in page 3 (line 65 onwards)
- Text added (line 143)